# Nitric Oxide Induced by Ammonium/Nitrate Ratio Ameliorates Low-Light Stress in *Brassica pekinesis*: Regulation of Photosynthesis and Root Architecture

**DOI:** 10.3390/ijms24087271

**Published:** 2023-04-14

**Authors:** Linli Hu, Xueqin Gao, Yutong Li, Jian Lyu, Xuemei Xiao, Guobin Zhang, Jihua Yu

**Affiliations:** 1Gansu Provincial Key Laboratory of Aridland Crop Science, Gansu Agricultural University, Lanzhou 730070, China; hull@gsau.edu.cn; 2College of Horticulture, Gansu Agricultural University, Lanzhou 730070, China; gaoxq@st.gsau.edu.cn (X.G.); liyutong115@163.com (Y.L.); lvjian@gsau.edu.cn (J.L.); xiaoxm@gsau.edu.cn (X.X.); zhanggb@gsau.edu.cn (G.Z.)

**Keywords:** ammonium: nitrate ratio, nitric oxide, photosynthesis, low light, *Brassica pekinesis*

## Abstract

Low-light intensity affects plant growth and development and, finally, causes a decrease in yield and quality. There is a need for improved cropping strategies to solve the problem. We previously demonstrated that moderate ammonium:nitrate ratio (NH_4_^+^:NO_3_^−^) mitigated the adverse effect caused by low-light stress, although the mechanism behind this alleviation is unclear. The hypothesis that the synthesis of nitric oxide (NO) induced by moderate NH_4_^+^:NO_3_^−^ (10:90) involved in regulating photosynthesis and root architecture of *Brassica pekinesis* subjected to low-light intensity was proposed. To prove the hypothesis, a number of hydroponic experiments were conducted. The results showed that in plants exposed to low-light intensity, the exogenous donors NO (SNP) and NH_4_^+^:NO_3_^−^ (N, 10:90) treatments significantly increased leaf area, growth range, and root fresh weight compared with nitrate treatment. However, the application of hemoglobin (Hb, NO scavenger), N-nitro-l-arginine methyl ester (L-NAME, NOS inhibitor), and sodium azide (NaN_3_, NR inhibitor) in N solution remarkably decreased the leaf area, canopy spread, the biomass of shoot and root, the surface area, and volume and tips of the root. The application of N solution and exogenous SNP significantly enhanced Pn (Net photosynthetic rate) and rETR (relative electron transport rates) compared with solo nitrate. While all these effects of N and SNP on photosynthesis, such as Pn, Fv/Fm (maximum quantum yield of PSII), Y(II) (actual photosynthetic efficiency), qP (photochemical quenching), and rETR were reversed when the application of Hb, L-NAME, and NaN_3_ in N solution. The results also showed that the N and SNP treatments were more conducive to maintaining cell morphology, chloroplast structure, and a higher degree of grana stacking of low-light treated plants. Moreover, the application of N significantly increased the NOS and NR activities, and the NO levels in the leaves and roots of mini Chinese cabbage seedlings treated with N were significantly higher than those in nitrate-treated plants. In conclusion, the results of this study showed that NO synthesis induced by the appropriate ammonia–nitrate ratio (NH_4_^+^:NO_3_^−^ = 10:90) was involved in the regulation of photosynthesis and root structure of *Brassica pekinesis* under low-light stress, effectively alleviating low-light stress and contributing to the growth of mini Chinese cabbage under low-light stress.

## 1. Introduction

Light conditions and nitrogen nutrition are two main factors that affect the yield of vegetables. Low light has a great influence on vegetable performance during the winter and spring seasons, as it affects various metabolic processes, including photosynthetic efficiency, which decreases vegetable crop production [1]. To mitigate the negative effects of low-light intensity stress on plants, several measures are being practiced these days. The application of a variety of exogenous plant growth regulators (PGRs), such as calcium and 5-aminolevulinic acid, has been used to improve low-light stress tolerance in plants [2,3]. In addition, supplementing LED light source can effectively alleviate low-light stress [4]. Nitrogen, as a mineral element most needed for the plant, is a key regulator of crop growth and development. The two main inorganic nitrogen forms that can be utilized by crops are ammonium (NH_4_^+^) and nitrate (NO_3_^−^) [5]. Previously, we have demonstrated that the application of an appropriate NH_4_^+^:NO_3_^−^ ratio could mitigate low-light stress in mini Chinese cabbage seedlings [6]. This result would be, subsequently, a new cropping strategy to address the problem of the low-light stress on the performance of vegetable crops. Therefore, it is very important to adequately understand the mechanism involved in the alleviation process.

In plants, nitric oxide (NO) is known as a small but important multifunctional bioactive signaling molecule and plays a critical function as a regulator of plant growth and development, including germination [7], root growth [8], flowering [9], and senescence [10]. Besides its function in the regulation of plant developmental processes, NO has been found to regulate plant response to stresses caused by heavy metal toxicity [11,12], water deficit [13,14], low-light intensity [15], salt stress [16], paraquat toxicity [17], and plant pathogen [18]. Under stress conditions, NO is known to be a stress regulator involved in signal transduction pathways. Although the formation of NO in plant cells is still the subject of continuous research, approximately two main oxidative and reductive pathways have been thought to be involved in NO biosynthesis in plants [19]. The oxidative pathways included oxidation of L-arginine by NOS-like enzyme, hydroxylamine, and polyamines, while the reductive pathways included enzymatic reduction of nitrite, including nitrate reductase (NR), xanthine oxidoreductase (XOR), plasma membrane-bound nitrite: NO reductase (PMNiNOR), cytochrome-c oxidase and/or reductase, and non-enzymatic reduction of nitrite under acidic or anoxic conditions [20]. It has also been found that NO can function as an antioxidant, quenching ROS during oxidative stress, reducing lipid peroxidation and regulating stomata opening and then mediating photosynthesis [21], as well as triggering calcium signaling and, therefore, improving the tolerance of plants to stressful conditions [22]. Based on the biological function and synthesis pathways of NO in plants, the hypothesis was proposed that NO was involved in the regulation of the photosynthesis and root architecture induced by the moderate NH_4_^+^:NO_3_^−^ ratio to decrease the negative effects of low light.

Mini Chinese cabbage (*Brassica pekinensis*), one of the most important Brassica vegetables worldwide, has remarkable economic benefits [23,24,25]. It has great health benefits, including anticancer, anti-obesity, and antioxidant effects [26]. When it is cultivated in greenhouses during winter, the low light is a key stress factor inhibiting plant growth and crop yield. Our research previously found that the adjustment of the NH_4_^+^:NO_3_^−^ ratio in the nutrient solution based on the light intensity is critical. However, the mechanism is still not very clear. In this paper, we hypothesized that the appropriate NH_4_^+^:NO_3_^−^ ratio would change the NO level and then trigger signal transduction pathways to regulate the photosynthesis and root architecture in *Brassica pekinesis* subjected to low-light intensity. Therefore, the aims of the current study were to explore the role of NO (applied exogenous nitric oxide donor or inhibited endogenous NO in the appropriate NH_4_^+^:NO_3_^−^ ratio nutrition) in regulating low-light intensity-induced root architecture and photosynthesis in mini Chinese cabbage and also the interaction between appropriate NH_4_^+^:NO_3_^−^ ratio (10:90) and NO in these processes.

## 2. Results

### 2.1. Morphyological Parameters and Biomass

To confirm the alleviation role of exogenous NO and endogenous NO, which resulted from moderate NH_4_^+^:NO_3_^−^ (10:90) on the low-light stress-induced growth inhibition, NO donor (SNP), NO biosynthesis inhibitors (L-NAME and NaN_3_), and a special NO scavenger (Hb) added into appropriate N solution were tested. We found that under low-light stress, compared with nitrate-treated plants, the leaf area, canopy spread, and root fresh weight of moderate N-treated plants remarkably increased by 26.3%, 50.3%, and 87.4%, respectively, and those of exogenous NO-treated plants remarkably increased by 22.2%, 35.1%, and 74.8%, respectively. Moreover, moderate N and exogenous NO also increased the shoot fresh weight and shoot and root dry weight, suggesting that exogenous NO can also mitigate the low-light intensity stress (Table 1). However, the application of L-NAME, NaN_3,_ and Hb in N solution significantly decreased the leaf area, leaf number, canopy spread, shoot and root fresh weight, and shoot dry weight, suggesting that the mitigation role of N on low-light intensity was reversed (Table 1). The phenotype of the differential treated seedlings showed the same results (Figure 1). Therefore, we partially concluded that both exogenous NO and appropriate N alleviated low-light intensity stress, and NO participated in the alleviation role of appropriate N on low-light stress in *Brassica pekinesis* seedlings.

### 2.2. Root Morphology Parameters

The root morphology parameters were also investigated, and the root photographs were obtained to analyze root morphological parameters (Figure 2 and Table 2). Under low-light stress, the seedlings treated with exogenous NO had 71% higher total root length, 129.4% higher root volume, and 176.7% higher root tips than those of seedlings treated with CK, respectively. There is no significant difference between exogenous NO and N treatments except for root tips. However, the application of Hb, L-NAME, and NaN_3_ in N solution significantly decreased the total length, surface area, volume, and tips of the root when compared with the N treatment. Thus, we partially concluded that both exogenous NO and appropriate N alleviated low-light intensity stress, and NO participated in the alleviation role of appropriate N on mini Chinese cabbage seedlings subjected to low-light intensity stress by regulating the root architecture.

### 2.3. Gas Exchange Parameters

The gas exchange parameters were further investigated to explore the mechanism of the phenomena. The Pn of leaves treated with N and SNP were 43.7% and 49.7% higher than those treated with solo nitrate, respectively. Moreover, when compared with the N treatment, the application of Hb, NaN_3_, and L-NAME in N solutions significantly decreased the Pn by 81%, 74%, and 58%, respectively. The Gs (Stomata conductance) and Tr (Transpiration rate) in plants treated with N and SNP were higher than those plants treated with solo nitrate. However, the decreased values of Gs and Tr were observed in N + Hb, N + NaN_3_, and N + L-NAME treatments. As for Ci (Intercellular CO_2_ concentration), there was no remarkable difference among N, SNP, and CK treatments, but the value of Ci was significantly increased in N + Hb, N + NaN_3_, and N + L-NAME treatments when compared with N treatment. All these data indicated that exogenous NO and endogenous NO induced by moderate NH_4_^+^:NO_3_^−^ ratio were involved in the regulation of gas exchange parameters. Otherwise, we speculated that the possible reason for NO to enhance net photosynthetic rate mainly might be the regulation of non-stomatal factors (Table 3).

### 2.4. Chlorophyll Fluorescence Imaging

From Figure 3, the plants treated with N and SNP had 7.3% and 6% higher Fv/Fm, 18.5% and 13.1% higher Y(II), and 9.4% and 5.5% higher qP than those of plants treated with solo nitrate, respectively. However, when the Hb, NaN_3_, and L-NAME were added into N treatment solutions, the Y(II), Fv/Fm, and qP declined compared with N treatment, and the decrease in Y(II) reached a significant difference. On the contrary, the N and SNP treatments decreased the Y(NO) (Quantum yield of regulated energy dissipation in PS II) compared with CK, whereas the N + Hb, N + NaN_3_, and N + L-NAME treatments remarkably increased the Y(NO) compared with N solution treatment (Figure 3B). As for Y(NPQ) (Quantum yield of non-regulated energy dissipation in PS II), there was no significant difference among N, SNP, and CK treatments and also among N + Hb, N + NaN_3_, N + L-NAME, and N treatments (Figure 3B). The ETR of the leaf gradually increased and then kept steady with the increase in light intensity. Furthermore, the ETR of plants treated with N and SNP were higher than that of those treated with CK, whereas the addition of Hb, NaN_3_, and L-NAME notably declined the ETR values compared to the N treatment. All the data pointed to the fact that exogenous NO and endogenous NO induced by moderate NH_4_^+^:NO_3_^−^ ratio played a vital role in regulating the chlorophyll fluorescence imaging parameters.

### 2.5. Chloroplast Ultrastructure

The results of changes that occurred in the mesophyll cells and chloroplast ultrastructure are shown in Figure 4. The leaf of seedlings grown in N and SNP treatment solutions under low-light intensity had regular cell shape and typical chloroplast shape. Moreover, there were smoothly arrayed grana stacking and a certain amount and suitable size of starch grains (Figure 4(b1–b3,c1–c3)), while the cell of seedlings treated with solo nitrate (CK) solution shrunk slightly. However, cell shrinkage and plasmolysis occurred (the double sorrow shown in Figure 4(d3)) when seedlings were treated with N + Hb, N + NaN_3_, and N + L-NAME solutions, and the number of starch grains obviously decreased, except for N + L-NAME-treated leaf (Figure 4(d1–d3,e1–e3,f1–f3)). In the mesophyll cell of leaf treated with N + L-NAME solutions, several big size starch grains filled chloroplast and caused the shapes of chloroplasts to swell severely; the degree of grana stacking became loose, and the number of mitochondria increased (Figure 4(f1–f3)). For N + Hb and N + NaN_3_-treated seedlings, there were plenty of osmiophilic granules and fewer starch grains in the chloroplast compared to those treated with N solution (Figure 4(d1–d3,e1–e3)).

### 2.6. NOS and NR Activity

As shown in Figure 5, the NOS activity of plants treated with N was significantly higher than those treated with other treatments. There was no significant difference among SNP, N + NaN_3_, and CK. The application of Hb and L-NAME significantly decreased the NOS activity. The NR activity in plants treated with N and SNP was significantly higher than that in CK, while the application of NaN_3_, L-NAME, and Hb in the N solution significantly decreased the NR activity. The above results implied that both the NOS pathway and NR pathway participated in the production of NO in *Brassica pekinesis* under low-light stress.

### 2.7. NO Level in Leaf and Root

The NO levels in the leaf and root are shown in Figure 6. The NO level in SNP-treated plants is the highest. The NO level in the N-treated plant has a significant difference in the leaf but not in the root when compared with SNP-treated plants. Moreover, N treatment remarkably enhanced NO levels in leaf and root compared with CK, which increased by 59.3% and 32.6%, respectively. However, the application of NaN_3_, L-NAME, and Hb in N solution remarkably reduced the NO level in the leaf and root.

## 3. Discussion

The problem of low-light intensity is one of the main factors against increased yield and quality of vegetables grown in a greenhouse, especially in winter. Low-light intensity changed root architecture and reduced the plant’s ability to efficiently photosynthesize. NO is a critical regulator in triggering the plant responses to low-light intensity stress [27,28]. In our study, exogenous NO had the same positive effect as appropriate NH_4_^+^:NO_3_^−^ on canopy spread, leaf area, and the biomass distribution of mini Chinese cabbage when seedlings grew under low-light conditions in comparison to solo nitrate (Table 1). Whereas the positive effects of N solution on plant growth phenotypes were reversed when the NO scavenger (Hb) and synthetic inhibitor (NaN_3_ and L-NAME) were added to N solution (Table 1), implying that exogenous NO and endogenous NO induced by appropriate NH_4_^+^:NO_3_^−^ had the alleviation role on low-light stress in *Brassica pekinesis* seedling. It is known that the relationship between NO synthesis and nitrogen metabolism was reported by Dean and Harper [29], who indicated the involvement of nitrate reductase (NR) in NO synthesis. NR catalyzes the reduction of nitrate (NO_3_^−^) to NO_2_^−^, which is further reduced to ammonium by nitrite reductase before it is converted into amino acids [30]. However, in recent times, NO_2_^-^ has widely been considered an important substrate for NO synthesis in plants since it could also be reduced to NO [31,32]. The NO levels in the leaf and root of plants treated with N and SNP were significantly higher than those treated with solo nitrate (Figure 5). Moreover, the NR and NOS activity in the leaf and root of the plant treated with N were remarkably higher than those treated with solo nitrate (Figure 4); the application of Hb, NaN_3_, and L-NAME in N solution reversed the positive effects, suggesting that appropriate ammonium concentration in nutrient solution improved the activity of key enzymes in two enzymatic pathways for NO synthesis in plants and promoted more NO_2_^-^ converted into NO under low light. Rockel et al. [33] indicated that the levels of NO production resulting from the reduction of nitrate to nitrite (only 1% of NR activity) were greatly low under normal conditions. However, NR-dependent NO production sped up in plant cells when plants were exposed to acidic conditions or high concentrations of nitrite and low nitrate. In our study, the addition of ammonium will create an acidic root environment and a relatively low nitrate nutrient solution. This is probably the reason for the increased NO level in mini Chinese cabbage seedlings.

NO has also been extensively proven to be involved in root development [8,34,35]. Previous investigations reported that the application of NO donors to different plant species roots promotes the formation of adventitious and lateral roots, root hair development, and root tip elongation, emphasizing the function of NO in root organogenesis [36,37,38]. Given the above findings, it is no wonder that NO also emerges as a ubiquitous signaling molecule in roots in plants under stressful conditions. Much research evidence revealed that abiotic stresses had been shown to induce NO synthesis in root tissues; thereby, a set of responses was contributed to allowing plants to adapt to various stressful conditions. For instance, the inhibition of root elongation caused by the aluminum toxicity in Hibiscus moscheutos was mitigated by NO donor sodium nitroprusside (SNP) [39], cadmium toxicity affected NO homeostasis in the root of Pisum sativum and Arabidopsis thaliana [40,41]. An over-accumulation of H_2_O_2_ and O_2_^-^ and a decrease in NO levels in primary and lateral roots were observed in pea seedlings treated with CdCl_2_, thereby causing a decrease in the length and number of lateral roots and changes in the structure of the principal roots affecting the xylem vessels, while an increase in NO accumulation in roots was observed after 24 h of treatment with cadmium in pea seedlings [42]. The difference could be ascribed to the cell response to short and long periods of cadmium treatment. In the present study, significantly higher NO levels were observed in the roots of N and SNP-treated plants under low-light intensity; meanwhile, we observed better root architecture in N and SNP-treated plants, which had longer total root length, larger surface area, the volume of the root, and more root tips. However, all the above positive effects of N treatment on root were reversed by the application of Hb, NaN_3_, and L-NAME, suggesting that endogenous NO induced by moderate NH_4_^+^/NO_3_^−^ participated in the regulation of root architecture in *Brassica pekinesis* under low-light stress.

Recent studies have shown that exogenous NO shows complex effects on plant leaf photosynthesis [43]. It had been demonstrated that exogenous NO suppressed the net photosynthetic rate of intact leaves of oat (*Avena sativa* L.) and alfalfa (*Medicago sativa* L.) at concentrations below those required to cause obvious injury [44]. However, SNP treatment increased the contents of total Chl, Chl a/b ratio, and chloroplasts Hill reaction activity in Cd-treated leaves of Brassica napus [45]. Similarly, NO also had a positive effect on photosynthesis under other abiotic stresses, such as salt, drought, low-light intensity, and heavy metals [12,15,16,46]. In the present study, the application of exogenous NO and moderate NH_4_^+^:NO_3_^−^ significantly enhanced Pn compared with the solo nitrate treatment, while the application of Hb, NaN_3_, and L-NAME reversed the positive effect of N on photosynthesis; moreover, the changing trend of stomatal conductance was similar to Pn, but the trend of Ci was opposite to Pn, suggesting that a reduction in photosynthetic capacity was attributed to not only stomatal limitation but also non-stomatal factors under low-light conditions. The decrease triggered by the low-light intensity in Pn was probably attributed to the degradation and disruption of chloroplasts and the consequent reduction in PSII efficiency. We then evaluated the changes in chloroplast ultrastructure and chlorophyll fluorescence parameters under various treatments. Low-light stress visibly damaged the organelles structure of leaves, which was consistent with the results of the study of Chinese cabbage ‘Biyu’ (salt-sensitive–alkali-tolerant) under saline–alkali stress [47], while the application of moderate NH_4_^+^/NO_3_^−^ or NO significantly maintained cell normal structure and improved the internal ultrastructure of mesophyll cells. However, the positive effects of NO and N on chloroplast ultrastructure were reversed by the application of Hb, NaN_3_, and L-NAME in N solution (Figure 3). Therefore, NO induced by moderate NH_4_^+^/NO_3_^−^ may mitigate the negative effects of low light on chloroplast ultrastructure. The ultrastructural change was similar to that reported in marigolds subjected to NO and H_2_O_2_ treatment when exposed to drought stress [48]. Chlorophyll fluorescence parameters reflect photosystems II (PS II) efficiency. We found that N and SNP treatments increased Fv/Fm, qP, Y(II), and rETR when compared with nitrate treatment, but decreased Y(NO), whereas the application of Hb, NaN_3_, and L-NAME in N reversed the effect of N on these parameters (Figure 2). It was suggested that endogenous NO induced by moderate NH_4_^+^/NO_3_^−^ might promote more light energy conversion to Y(II), enhancing photochemical energy distribution, although Y(NPQ) remained stable. These findings were similar to the report on mustard (Brassica juncea. L cv. RGN-48), where PSII activity was enhanced by the foliar spray of NO in the salt-stressed plants [49]. In the end, a model illustrating how an appropriate ammonium/nitrate ratio (10:90) alleviates low-light intensity stress in mini Chinese cabbage seedlings was proposed (Figure 6), which shows that the addition of suitable ammonium concentration in nitrate solution increased the activities of NOS and NR enzymes which promoted endogenous NO synthesis, thus improving the chloroplast ultrastructure, PSII activity, and root architecture, and finally enhanced the low-light tolerance in *Brassica pekinesis*.

## 4. Material and Methods

### 4.1. Plant Material and Experiment Design

Mini Chinese cabbage (*Brassica pekinesis*) seeds were obtained from a certified local agro-input dealer in the Anning district, Lanzhou, Gansu province of China. The uniform-sized seeds were disinfected with 0.1% (*w*/*v*) potassium permanganate for 5 min and then washed well with distilled water. After sterilization and washing, seeds were placed on double-layer moist filter paper and kept at 25± 2 °C in the dark for 16 h for germination. After that, the germinated seeds were sown in quartz sand, disinfected with carbendazim, and spray-irrigated with a quarter of Hoagland’s nutrient solution once a day. When the second leaf fully expanded, groups of 4 seedlings with uniform size were transferred to 300 mL black plastic containers (the diameter on the top was 12 cm, the diameter on the bottom was 9 cm, and the height was 4 cm) filled with distilled water for 1 d, and then the distilled water was changed to different treatment solutions. The treatment codes and the corresponding ingredients of treatment solutions were listed as follows: (1) CK, ammonium/nitrate = 0/100 and total nitrogen concentration was 5 mmol·L^−1^; (2) N, ammonium/nitrate = 10/90 and total nitrogen concentration was 5 mM; (3) SNP, 100 μmol·L^−1^ sodium nitroprusside; (4) N + Hb, the addition of 0.1% hemoglobin (Hb) into N solution to remove NO from plants; (5) N + NaN_3_, the addition of 50 μmol·L^−1^ sodium azide (NaN_3_) into N solution to inhibit NR activity from reducing NO synthesis; (6) N + L-NAME, the addition of 25 μmol·L^−1^ Nω-nitro-L-arginine methyl ester (L-NAME) into N solution to inhibit NOS activity from reducing NO synthesis. The composition of the CK and N nutrition solution was referred to in the previous experiment [6].

The nitrification inhibitor (Dicyandiamide, 7 μmol·L^−1^) was added to each container to suppress the transformation of nitrate to ammonium. The initial pH of every container was adjusted to 6.5–7.0 by applying 0.1 mol· L^−1^ HCl or 0.1 mol·L^−1^ NaOH. The seedlings were placed in an artificial climate chamber with a photoperiod of 12 h, light intensity of about 100 μmol·m^−2^ s^−1^ (the low-light intensity is always between 85 and 150 μmol m^−2^ s^−1^ in the day during winter in the greenhouse), day and night temperature of 25 ± 1 °C and 18 ± 1 °C. The treatment solutions were completely renewed at 2-day intervals. The containers, which contained four seedlings each, were arranged in the artificial climate chamber using a completely randomized design.

### 4.2. Evaluation of Morphological Parameters and Biomass

Eight days after seedlings were exposed to different treatment solutions, canopy spread, the third leaf area, leaf number, and biomass of mini Chinese cabbage seedlings were evaluated following the method of Hu et al. [50] with a slight modification. The leaf of mini Chinese cabbage was deemed as elliptical, and, therefore, the third leaf area of seedlings was calculated as follows:Leaf area = k × π × A × B/4 (k = 1.10)(1)
where A is the length of the leaf, and B is the width of the leaf. The coefficient k was calculated by the ratio of actual area to approximate area for the same leaf. The leaf number of which leaf area exceeded 1 mm^2^ was counted. When harvested, the seedling was divided into two parts, shoot and root. The shoot and root fresh weights were recorded separately. In the following, the samples were placed into an oven at 105℃ for 15 min and, afterward, at 80 ℃ until constant weight. The shoot and root dry weights were evaluated by an electronic balance (degree of accuracy: 0.001 g).

### 4.3. Analysis of Root Morphology Parameters

The root morphology images obtained by a root system scanner (Expression 11000XL, EPSON) were analyzed by the WinRHIZO software (Regent Instruments Inc., Quebec, G1V 1V4, Canada). Subsequently, the data of total root length, surface area, average diameter, root volume, and root tips of each root sample were obtained, and the means of these data were calculated by SPSS 22.0 software.

### 4.4. Determination of Gas Exchange Parameters

Gas exchange parameters of the fourth leaf, including the net photosynthetic rate (Pn), stomata conductance (Gs), intercellular CO_2_ concentration (Ci), and transpiration rate (Tr), were determined with a portable photosynthetic system (CIRAS-2, PP System, Hertfordshire, UK). In the measurement process, parameters were set as follows: the leaf temperature was 25 °C, the PPFD was 1000 mmol·m^−2^·s^−1^, ambient CO_2_ concentration was about 380 mmol·mol^−1^, and the relative humidity was 75%.

### 4.5. Measurement of Chlorophyll Fluorescence Imaging

Images and parameters of chlorophyll fluorescence were obtained from the fourth leaf of each mini Chinese cabbage seedling with the MAXI version of the Imaging-PAM fluorometer and the Imaging Win software (Walz, Effeltrich, Germany). The seedlings were initially kept in darkness for 60 min to completely open the photosystems II (PSII) reaction centers, after which the leaf was individually fixed in support and then exposed to a weak, modulated measuring light (0.5 μmol·m^−2^ s^−1^) to obtain the initial fluorescence (Fo) when the PSII reaction centers were “open.” Subsequently, a saturation pulse (2700 μmol·m^−2^·s^−1^) was given to determine the maximum fluorescence (Fm) for 0.8 s when the PSII reaction centers were expected to be “closed.” In the following, the leaf was exposed to actinic light of 81 μmol·m^−2^· s^−1^ for 5 min to acquire the steady-state fluorescence (Fs). The light-adapted maximum fluorescence (Fm’) was achieved after applying a saturation pulse of 2700 μmol·m^−2^·s^−1^. Light-adapted minimal fluorescence (Fo’) was obtained when the actinic light was turned off in the presence of far-red light. Finally, the relative electron transport rates (rETR) at a given photosynthetic active radiation actinic irradiance (PAR = 0, 21, 56, 111, 186, 281, 336, 396, 461, 531, 611,701, 926, 1251 μmol·m^−2^·s^−1^) were obtained, as described by White and Critchley [51]. The following chlorophyll fluorescence parameters were calculated as the following computational formulas using the Imaging Win software [52]:Y(II) = (Fm’ − Fs)/Fm’(2)
Y(NO) = 1/(NPQ + 1 + qL(Fm/Fo − 1))(3)
Y(NPQ)= 1 − Y(II) − 1/(NPQ + 1 + qL(Fm/Fo − 1))(4)
Fv/Fm = (Fm − Fo)/Fm(5)
qP = (Fm’ − F)/(Fm’ − Fo’)(6)
rETR = PAR × Y(II) × 0.84 × 0.5(7)

### 4.6. Observation of Chloroplast Ultrastructure

The chloroplast ultrastructure section and observation of mini Chinese cabbage leaf tissue were carried out according to the method of Hu et al. (2015) [6].

### 4.7. Assay of the Nitric Oxide Synthase (NOS) and Nitrate Reductase (NR) Activity

The activities of NOS and NR in leaves were measured using the NOS ELISA Kit (Shanghai Baiye Biotechnology Center, Shanghai, China) and the NR determination Kit (Suzhou Keming Biotechnology Co., Suzhou, China) according to the manufacturer’s instructions. The OD values of NOS and NR were obtained at 450 nm and 540 nm, respectively. The protein concentration of each extraction was analyzed by the method of Bradford [53]. Finally, data of NOS and NR activity were expressed as U·mg^−1^ protein.

### 4.8. Assay of the NO Level in Leaf and Root

The Griess reagent method described by Hu et al. [54] was used to measure the NO level in the leaf and root of mini Chinese cabbage with some modifications. The samples of leaf and root (0.5 g each) were ground into homogenates in 3 mL 50 mmol·L^−1^ ice-cold acetic acid buffer containing 4% zinc diacetate (pH 3.6). The homogenates were centrifuged at 4 °C at 1000× *g* for 15 min and then collected the supernatants. The above buffer (1 mL) was added to the residue, the above process was repeated, and then the three-time supernatants of each sample were mixed together. After mixing, we added 0.1 g activated charcoal, swirled and filtered, and then collected the filtrate as the reaction solution. We mixed 2 mL Griess reagent (1% p-aminobenzenesulfonic acid, 0.1% N-naphthyl-ethylenediamine, 5% phosphoric acid) and 2 mL reaction solution together and incubated these at room temperature for 30 min. The absorbance was read at 540 nm, and the levels of NO in the leaf and root were calculated by relating it with a standard curve of NaNO_2_.

### 4.9. Data Analysis

All the data collected were subjected to analysis of variance, and the treatment means were compared with SPSS 16.0 software (SPSS Institute Inc., Chicago, IL, USA) according to Tukey’s test (*p* ≤ 0.05). Each experiment was performed with three replications, and the results were presented as mean ± SE.

## 5. Conclusions

We concluded that the acquisition of tolerance to low-light stress in *Brassica pekinesis* is owed to improved root architecture and significantly increased photosynthetic performance. As a stress-responded plant signal molecule, NO protects *Brassica pekinesis* against low-light stress by regulating root architecture, maintaining intact cell and chloroplast structure, and enhancing the photosynthesis rate. Appropriate NH_4_^+^:NO_3_^−^ promoted NO release by enhancing the activities of NR and NOS enzymes. Under low-light conditions, NO might operate downstream of appropriate NH_4_^+^:NO_3_^−^ in mini Chinese cabbage seedlings (Figure 7).

## Figures and Tables

**Figure 1 ijms-24-07271-f001:**
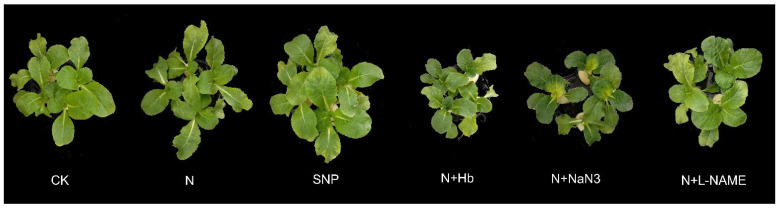
Effects of different ammonium/nitrate ratios, exogenous nitric oxide, and the addition of NO scavenger and biosynthesis inhibitors in ammonium/nitrate solution on shoot and root morphology of mini Chinese cabbage seedling exposed to low-light stress. Shoot morphology of 8-day-old mini Chinese cabbage seedlings cultivated in hydroponics system and fertilized with different treated solutions: CK, ammonium/nitrate = 0/100, and the total nitrogen concentration was 5 mM; N, ammonium/nitrate = 10/90, and the total nitrogen concentration was 5 mM; SNP, 100 μmol/L sodium nitroprusside (SNP); N + Hb, 0.1% hemoglobin (Hb) was added to N solution; N + NaN_3_, 50 μmol/L sodium azide (NaN_3_) was added to N solution; N + L-NAME, 25 μmol/L Nω-nitro-L-arginine methyl ester (L-NAME) was added to N solution.

**Figure 2 ijms-24-07271-f002:**
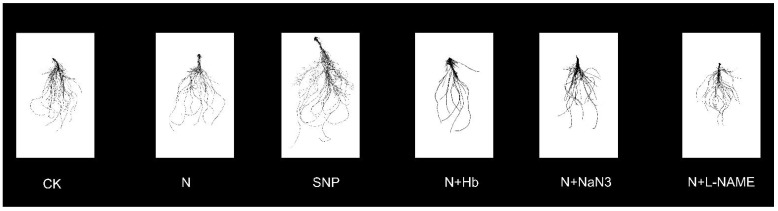
Effects of different ammonium/nitrate ratios, exogenous nitric oxide, and the addition of NO scavenger and biosynthesis inhibitors in ammonium/nitrate solution on shoot and root morphology of mini Chinese cabbage seedling exposed to low-light stress. Root morphology of 8-day-old mini Chinese cabbage seedlings cultivated in hydroponics system and fertilized with the above-treated solutions. The photographs were obtained by scanning roots with root system scanner (Expression 11000XL, EPSON) and used to analyze root morphological parameters by software Win RHIZO version 5.0 (Regent Instruments, Inc., Quebec, G1V 1V4, Canada).

**Figure 3 ijms-24-07271-f003:**
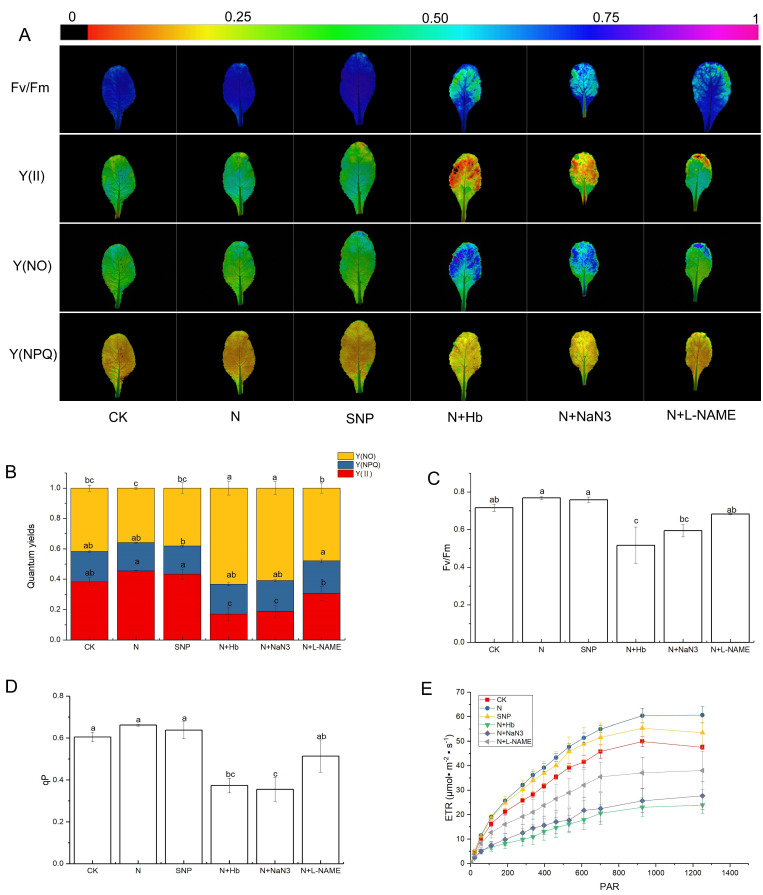
The chlorophyll fluorescence parameters of mini Chinese cabbage seedlings exposed to low−light stress. (**A**) Images of maximal PSII quantum efficiency (Fv/Fm), effective PSII quantum yield Y(II), quantum yield of non−regulated energy dissipation Y(NO), and quantum yield of regulated energy dissipation Y(NPQ) determined on the third leaf of mini Chinese cabbage seedling fertilized with ammonium/nitrate = 0/100 (CK), ammonium/nitrate = 10/90 (N), sodium nitroprusside (SNP), ammonium/nitrate (10/90)+ hemoglobin (N + Hb), ammonium/nitrate (10/90)+ sodium azide (N + NaN_3_), ammonium/nitrate (10/90) + Nω-nitro-L-arginine methyl ester (N + L-NAME). (**B**) Data of Y(II), Y(NO), and Y(NPQ). (**C**) Data of Fv/Fm. (**D**) Data of qP. (**E**) Data of ETR. Means from three leaves followed by different letters were significantly different (*p* ≤ 0.05) according to Tukey’s test. Bars indicated the standard errors of the means (*n* = 3).

**Figure 4 ijms-24-07271-f004:**
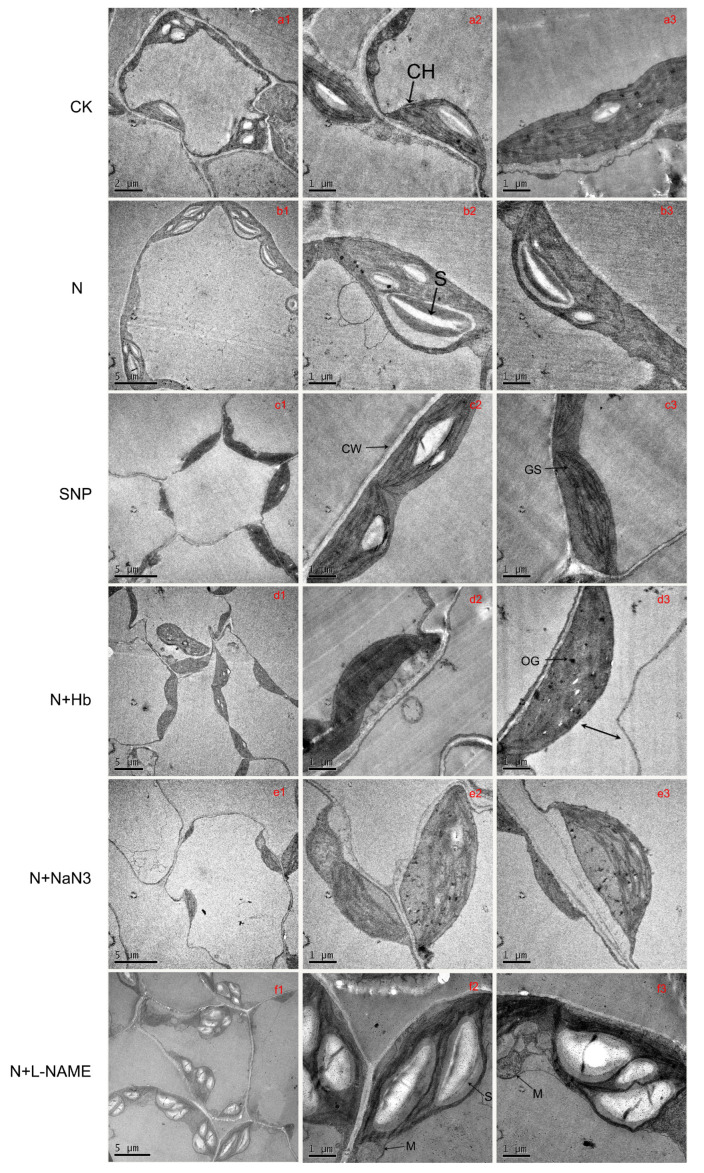
Effects of different ammonium/nitrate ratios, exogenous nitric oxide, and the addition of NO scavenger and biosynthesis inhibitors in ammonium/nitrate solution on chloroplast ultrastructure of mini Chinese cabbage leaves exposed to low-light stress, fertilized with the following solutions: CK (**a1**–**a3**); N (**b1**–**b3**); SNP (**c1**–**c3**); N + Hb (**d1**–**d3**); N + NaN_3_ (**e1**–**e3**); N + L-NAME (**f1**–**f3**). Abbreviations: CH, chloroplast; CW, cell wall; OG, osmiophilic globules; GS, grana stacking; S, starch; M, mitochondria.

**Figure 5 ijms-24-07271-f005:**
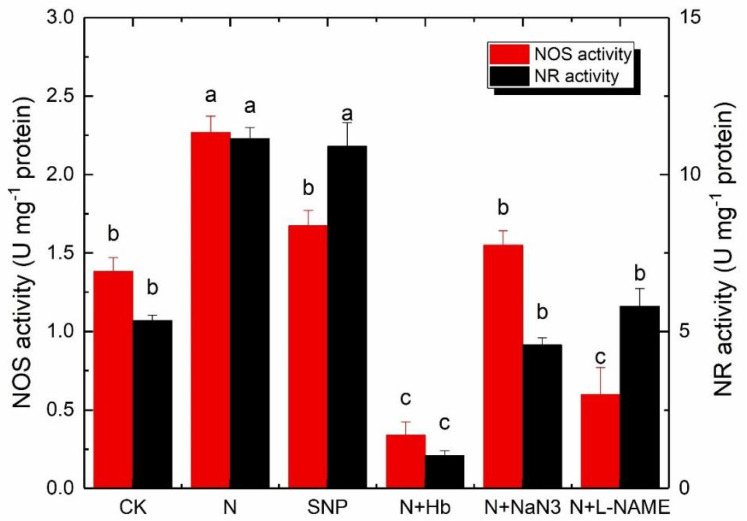
Effects of different ammonium/nitrate ratios, exogenous nitric oxide, and the addition of NO scavenger and biosynthesis inhibitors in ammonium/nitrate solution on NOS and NR activities of mini Chinese cabbage leaves exposed to low-light stress. Data represented means, and bars indicated the standard errors of the means (*n* = 3). Significant differences (*p* ≤ 0.05) between treatments were indicated by different letters, according to Tukey’s test.

**Figure 6 ijms-24-07271-f006:**
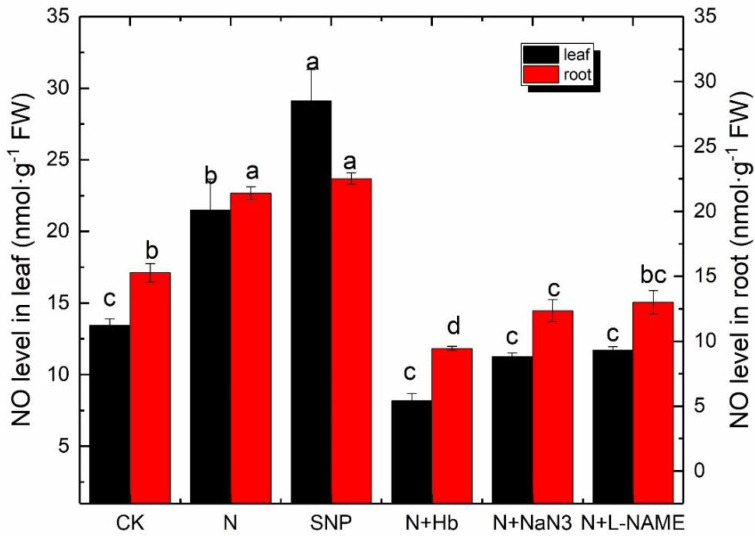
Effects of different ammonium/nitrate ratios, exogenous nitric oxide, and the addition of NO scavenger and biosynthesis inhibitors in ammonium/nitrate solution on NO levels of leaf and root in mini Chinese cabbage exposed to low−light stress. Data represented means, and bars indicated the standard errors of the means (*n* = 3). Significant differences (*p* ≤ 0.05) between treatments were indicated by different letters, according to Tukey’s test.

**Figure 7 ijms-24-07271-f007:**
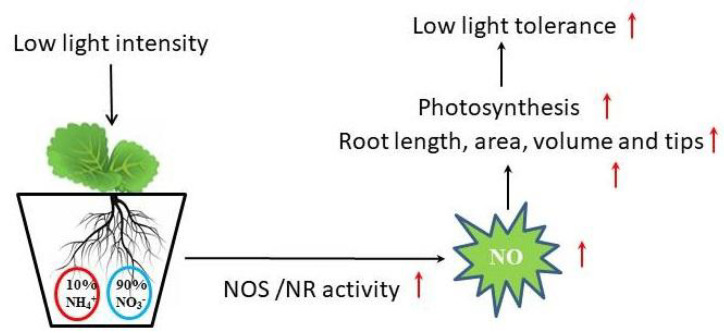
A proposed model illustrating how appropriate ammonium/nitrate ratio (10:90) alleviates low-light intensity stress in mini Chinese cabbage seedlings.

**Table 1 ijms-24-07271-t001:** Effects of exogenous nitric oxide, different ammonium/nitrate ratios, and the application of NO scavenger and biosynthesis inhibitors in ammonium/nitrate solution on growth parameters and biomass in *Brassica pekinesis* exposed to low-light stress.

Treatment	Leaf Area (cm^2^)	Canopy Spread (cm^2^)	Leaf Number (No·plant^−1^)	Fresh Weight of Shoot (g)	Fresh Weight of Root (g)	Dry Weight of Shoot (g)	Dry Weight of Root (g)
CK	19.17 ± 1.12 b *	99.95 ± 5.25 b	5.8 ± 0.13 abc	2.23 ± 0.18 ab	0.103 ± 0.006 b	0.133 ± 0.005 ab	0.008 ± 0.003 ab
N	24.21 ± 1.70 a	150.24 ± 9.03 a	6.2 ± 0.20 a	2.60 ± 0.47 a	0.193 ± 0.025 a	0.183 ± 0.013 a	0.009 ± 0.001 a
SNP	23.42 ± 1.86 a	135.02 ± 6.11 a	5.9 ± 0.23 ab	2.53 ± 0.34 a	0.180 ± 0.005 a	0.164 ± 0.019 a	0.009 ± 0.003 a
N + Hb	14.01 ± 0.55 c	78.2 ± 5.09 c	5.3 ± 0.15 c	1.50 ± 0.30 b	0.032 ± 0.004 c	0.102 ± 0.008 b	0.001 ± 0 b
N + NaN_3_	15.09 ± 0.96 bc	85.5 ± 7.02 bc	5.3 ± 0.15 c	1.47 ± 0.12 b	0.057 ± 0.009 c	0.107 ± 0.027 b	0.006 ± 0.002 ab
N + L-NAME	15.86 ± 1.67 bc	96.49 ± 8.00 bc	5.5 ± 0.17 bc	1.47 ± 0.12 b	0.060 ± 0.017 c	0.109 ± 0.007 b	0.005 ± 0.002 ab

***** Values were presented with means ± SE (*n* = 3 with 10 seedlings per replication), and different letters in the same column indicated significant differences (*p* ≤ 0.05) among treatments. Abbreviations: **CK**, ammonium/nitrate = 0/100, and the total nitrogen concentration was 5 mmol·L^−1^; **N**, ammonium/nitrate = 10/90, and the total nitrogen concentration was 5 mmol·L^−1^; **SNP**, 100 μmol·L^−1^ sodium nitroprusside (SNP); **N + Hb**, 0.1% hemoglobin (Hb) was added to N solution; **N + NaN_3_**, 50 μmol·L^−1^ sodium azide (NaN_3_) was added to N solution; **N + L-NAME**, 25 μmol·L^−1^ Nω-nitro-L-arginine methyl ester (L-NAME) was added to N solution.

**Table 2 ijms-24-07271-t002:** Effects of exogenous nitric oxide, different ammonium/nitrate ratios, and the application of NO scavenger and biosynthesis inhibitors in ammonium/nitrate solution on root morphological parameters in *Brassica pekinesis* exposed to low-light stress.

Treatment	Total Root Length (cm)	Surface Area (cm^2^)	Average Diameter (mm)	Root Volume (cm^3^)	Root Tips
CK	243.09 ± 3.85 bc *	25.14 ± 2.50 ab	0.25 ± 0.02 c	0.17 ± 0.01 bc	181.33 ± 1.86 bc
N	304.52 ± 31.03 ab	30.43 ± 1.21 a	0.30 ± 0.02 bc	0.28 ± 0.07 ab	230.67 ± 16.42 b
SNP	415.71 ± 96.33 a	31.37 ± 6.07 a	0.25 ± 0.01 c	0.39 ± 0.07 a	501.67 ± 91.63 a
N + Hb	125.52 ± 17.45 c	16.33 ± 2.31 b	0.55 ± 0.08 a	0.12 ± 0.01 c	63.33 ± 3.84 c
N + NaN_3_	145.58 ± 20.90 c	18.10 ± 3.98 b	0.43 ± 0.08 ab	0.13 ± 0.02 c	72.67 ± 12.57 c
N + L-NAME	182.80 ± 14.92 bc	16.52 ± 1.80 b	0.29 ± 0.03 bc	0.12 ± 0.03 c	109 ± 11.93 bc

***** Values (mean ± SE) were the averages of three independent experiments, and different letters in the same column indicated significant differences (*p* ≤ 0.05) among treatments.

**Table 3 ijms-24-07271-t003:** Effects of exogenous nitric oxide, different ammonium/nitrate ratios, and the application of NO scavenger and biosynthesis inhibitors in ammonium/nitrate solution on root photosynthetic parameters in *Brassica pekinesis* exposed to low-light stress.

Treatment	Pn (molCO_2_ m^−2^ s^−1^)	Gs (mmol·H_2_O m^−2^ s^−1^)	Tr (mmol·H_2_O m^−2^ s^−1^)	Ci (mol CO_2_ mol^−1^)
CK	5.03 ± 0.32 b	126 ± 19.89 bc	2.53 ± 0.28 abc	369 ± 11.53 bc
N	7.23 ± 0.38 a	186 ± 38.59 ab	3.33 ± 0.50 ab	360 ± 12.44 c
SNP	7.53 ± 0.64 a	217 ± 35.59 a	3.67 ± 0.47 a	368 ± 4.16 bc
N + Hb	1.37 ± 0.22 c	63 ± 3.93 c	1.30 ± 0.00 c	406 ± 4.33 ab
N + NaN3	1.90 ± 0.15 c	132 ± 22.36 bc	1.87 ± 0.46 c	422 ± 7.02 a
N + L-NAME	3.00 ± 1.21 c	120 ± 15.10 bc	2.27 ± 0.35 bc	408 ± 24.94 ab

Values (mean ± SE) were the averages of three independent experiments, and different letters in the same column indicated significant differences (*p* ≤ 0.05) among treatments.

## Data Availability

Not applicable.

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
