# Peer review of "Nitric Oxide Induced by Ammonium/Nitrate Ratio Ameliorates Low-Light Stress in Brassica pekinesis: Regulation of Photosynthesis and Root Architecture"

_ijms, 2023, doi:10.3390/ijms24087271_

Round 1

Reviewer 1 Report

Peer review on the article “Nitric oxide induced by ammonium/nitrate ratio ameliorates low light stress in Brassica pekinesis: Regulation of photosynthesis and root architecture.” written by Linli Hu, Xueqin Gao, Yutong Li, Jian Lyu, Xuemei Xiao, Guobin Zhang and Jihua.

This paper discusses the role of the synthesis of nitric oxide (NO) induced by moderate NH4+:NO3- (10:90) involved in regulating photosynthesis and root architecture of Brassica pekinesis subjected to low light intensity. It is well known that low light intensity affects plant growth and development, and causes low of yield and quality. Mini Chinese cabbage (Brassica pekinensis) is an important Brassica vegetable grown as a salad crop not only in China (Fordham - Hadley, 2003). It has great health benefits, including anticancer, anti-obesity, and antioxidant effects (Kim et al., 2018). So, even a minor improvement in its growing technology is worth to publish.

The main contribution of this paper is that NO synthesis induced by the application of appropriate NH4+:NO3-(10:90) involved in regulating photosynthesis and root architecture in Brassica pekinesis exposed to low light stress. Stress-responded plant signal molecule, NO protects Brassica pekinesis, mini Chinese cabbage seedlings against low light stress through regulating root architecture, maintaining intact cell and chloroplast structure and enhancing the photosynthesis rate. This paper helps to adequately understand the mechanism involved in the alleviation process.

I recommend that this paper be accepted.

Major comments.

The results obtained will be useful in growing and propagation of mini Chinese cabbage with possibly lower costs and less energy. But this needs a little bit further study. Research with supplemental LED lighting revealed higher amount of total polyphenol and total flavonoids in Chinese cabbage (Lee et al., 2016).

The statistical analysis of this paper is suitable to its objective.

Minor Comments.

It is advised to include into the next paper on this topic: Li et al., 2021. Also, an earlier  paper by Ducsay and Varga, (2003) is also seems to worth to mention at least in the introduction

References

Ducsay, L., Varga, L. (2003). Cultivation of Brassica pekinensis under different forms of nitrogen nutrition. Horticultural Science, Czech Academy of Agricultural Sciences, 30, 3, 112-115.

Fordham, R., Hadley, P. (2003): Vegetables of Temperate Climates | OrientalBrassicas. in Benjamin Caballero eds., Encyclopedia of Food Sciences and Nutrition (Second Edition), Academic Press, 5938-5941. ISBN9780122270550

He, Y. K., Xue, W. X., Sun, Y.D., Yu, X. H. and Liu, P. L. (2000): Leafy head formation of the progenies of transgenic plants of Chinese cabbage with exogenous auxin genes. Cell Research, 10, 151– 160.

Kim, H. W., Jang, J. J., Kim, N. H., Lee, N. Y., Cho, T. J., Kim, S. H., Rhee, M. S. (2018): Factors that determine the microbiological quality of ready-to-use salted napa cabbage (Brassica pekinensis): Season and distribution temperature. Food Control, 87, 1–8.

Lee, M. K., Arasu, M. V., Park, S., Byeon, D. H., Chung, S.-O., Park, S. U., Lim, Y.-P., Kim, S.-J., (2016). LED lights enhance metabolites and antioxidants in Chinese cabbage and kale. Brazilian Archives of Biology and Technology, 259, e16150546.

Li, N., Zhang, Z., Gao, S., Lv Y, Chen, Z., Cao, B., Xu, K. (2021): Different responses of two Chinese cabbage (Brassica rapa L. ssp. pekinensis) cultivars in photosynthetic characteristics and chloroplast ultrastructure to salt and alkali stress. Planta, 254, 5, 102.

Yu, X. H., Peng, J. S., Feng, X. Z., Yang, S. X., Zheng, Z. R., Tang, X. R. and He, Y. K. (2000): Cloning and structural and expressional characterization of BcpLH gene preferentially expressed in folding leaf of Chinese cabbage. Science, China Life Sciences, 43, 321– 329.

Author Response

Dear reviewer,

        Thank you for your helpful suggestions! After carefully reading these suggestions, we have modified the manuscript accordingly, and detailed corrections are listed below point by point. Now list them as following:

Response to Reviewer 1 Comments

Point 1: The results obtained will be useful in growing and propagation of mini Chinese cabbage with possibly lower costs and less energy. But this needs a little bit further study. Research with supplemental LED lighting revealed higher amount of total polyphenol and total flavonoids in Chinese cabbage (Lee et al., 2016).

The statistical analysis of this paper is suitable to its objective.

Response 1: Thanks for the reviewer's comments. In this experiment, seedlings with total nitrate nitrogen showed weak growth under low light stress. Using partial ammonium nitrogen instead of nitrate nitrogen to alleviate low light stress. Exogenous nitric oxide can also alleviate low light stress, and an appropriate proportion of ammonium nitrate is beneficial for the production of endogenous nitric oxide. This experiment demonstrates from a pharmacological perspective that an appropriate proportion of ammonium nitrate can alleviate low light stress by inducing the synthesis of nitric oxide.

Point 2: It is advised to include into the next paper on this topic: Li et al., 2021. Also, an earlier paper by Ducsay and Varga, (2003) is also seems worth to mention at least in the introduction.

References

Ducsay, L., Varga, L. (2003). Cultivation of Brassica pekinensis under different forms of nitrogen nutrition. Horticultural Science, Czech Academy of Agricultural Sciences, 30, 3, 112-115.

Fordham, R., Hadley, P. (2003): Vegetables of Temperate Climates | OrientalBrassicas. in Benjamin Caballero eds., Encyclopedia of Food Sciences and Nutrition (Second Edition), Academic Press, 5938-5941. ISBN9780122270550

He, Y. K., Xue, W. X., Sun, Y.D., Yu, X. H. and Liu, P. L. (2000): Leafy head formation of the progenies of transgenic plants of Chinese cabbage with exogenous auxin genes. Cell Research, 10, 151– 160.

Kim, H. W., Jang, J. J., Kim, N. H., Lee, N. Y., Cho, T. J., Kim, S. H., Rhee, M. S. (2018): Factors that determine the microbiological quality of ready-to-use salted napa cabbage (Brassica pekinensis): Season and distribution temperature. Food Control, 87, 1–8.

Lee, M. K., Arasu, M. V., Park, S., Byeon, D. H., Chung, S.-O., Park, S. U., Lim, Y.-P., Kim, S.-J., (2016). LED lights enhance metabolites and antioxidants in Chinese cabbage and kale. Brazilian Archives of Biology and Technology, 259, e16150546.

Li, N., Zhang, Z., Gao, S., Lv Y, Chen, Z., Cao, B., Xu, K. (2021): Different responses of two Chinese cabbage (Brassica rapa L. ssp. pekinensis) cultivars in photosynthetic characteristics and chloroplast ultrastructure to salt and alkali stress. Planta, 254, 5, 102.

Yu, X. H., Peng, J. S., Feng, X. Z., Yang, S. X., Zheng, Z. R., Tang, X. R. and He, Y. K. (2000): Cloning and structural and expressional characterization of BcpLH gene preferentially expressed in folding leaf of Chinese cabbage. Science, China Life Sciences, 43, 321– 329.

Response 2:

Thanks for the reviewer's comments. We have cited the following literature in the introduction and discussion section.

Ducsay, L., Varga, L. (2003). Cultivation of Brassica pekinensis under different forms of nitrogen nutrition. Horticultural Science, Czech Academy of Agricultural Sciences, 30, 3, 112-115. (Line: 49-50)

Fordham, R., Hadley, P. (2003): Vegetables of Temperate Climates | OrientalBrassicas. in Benjamin Caballero eds., Encyclopedia of Food Sciences and Nutrition (Second Edition), Academic Press, 5938-5941. ISBN9780122270550 (Line: 78-79)

He, Y. K., Xue, W. X., Sun, Y.D., Yu, X. H. and Liu, P. L. (2000): Leafy head formation of the progenies of transgenic plants of Chinese cabbage with exogenous auxin genes. Cell Research, 10, 151– 160. (Line: 78-79)

Kim, H. W., Jang, J. J., Kim, N. H., Lee, N. Y., Cho, T. J., Kim, S. H., Rhee, M. S. (2018): Factors that determine the microbiological quality of ready-to-use salted napa cabbage (Brassica pekinensis): Season and distribution temperature. Food Control, 87, 1–8. (Line: 79-80)

Lee, M. K., Arasu, M. V., Park, S., Byeon, D. H., Chung, S.-O., Park, S. U., Lim, Y.-P., Kim, S.-J., (2016). LED lights enhance metabolites and antioxidants in Chinese cabbage and kale. Brazilian Archives of Biology and Technology, 259, e16150546. (Line: 47-48)

Li, N., Zhang, Z., Gao, S., Lv Y, Chen, Z., Cao, B., Xu, K. (2021): Different responses of two Chinese cabbage (Brassica rapa L. ssp. pekinensis) cultivars in photosynthetic characteristics and chloroplast ultrastructure to salt and alkali stress. Planta, 254, 5, 102. (Line: 327-328)

Yu, X. H., Peng, J. S., Feng, X. Z., Yang, S. X., Zheng, Z. R., Tang, X. R. and He, Y. K. (2000): Cloning and structural and expressional characterization of BcpLH gene preferentially expressed in folding leaf of Chinese cabbage. Science, China Life Sciences, 43, 321– 329. (Line: 78-79)

Thank you again for your useful comments and suggestions on the improvement of our manuscript.

The manuscript has been resubmitted to your journal. We look forward to your positive response.

Yours sincerely,

Linli Hu

Reviewer 2 Report

Dear authors,

In this article the authors investigate how the inclusion of 10% ammonium improves the tolerance of plants to low light and that this process is mediated by the production of NO. The article is not written carefully, it must be thoroughly reviewed and corrected. Scientifically, the idea is good, but I am afraid that methodologically a series of controls are needed so that the hypothesis that it deals with can be credible. In this paper, there are too many variables without proper controls. Authors must clearly answer these questions for this article to be published.

Majors:

*L31-34: It seems that the sentence is not finished, it is not understood, please rewrite it

*L66:” (PMNiNOR)” what is this ,did you mean NOFNiR? Clarify it please

*I do not understand why the experiment of Hb, NaN3 and L-NAME alone (that is, without the N condition) was not carried out to compare them with the CK control, as it was done with SNP. Without those controls, how do they know that the effect of such molecules is due to the NO produced under the N conditions? if they don't have a control to compare? I do not get it.

*I also don't understand why the N+SNP treatment was not done? to see if there is a synergistic effect of both or not. if there were a synergistic effect, each one would be producing different effects, and this has not been tested, and should have been done.

*Fig 3E, the statistical study is missing.

*Figure 4. The white letters that indicate the particular conditions of each photo cannot be seen, nor can the scale.

*”When naming Fig 5, they make a mistake and name it as 4”. and from there all the figures are badly named. Surely a figure was removed shortly before, denotes that due care has not been taken in sending the latest version.

*L287: “In our study, the addition of ammonium will create an acidic root environment and a relative low nitrate nutrient solution. This is probably the reason for increased NO level in mini Chinese cabbage seedling.” What I understand is that the ammonia would produce the opposite, raise the pH. Have you quantified the pH? Is there any bibliographical support for this hypothesis?. L375: “The initial pH of every container was adjusted to 6.5–7.0 via applying 0.1 mol· L−1 HCl or 0.1 mol·L−1 NaOH”  How does the hypothesis of pH variation match this fact?

Minors:

*L19: " the exogenous NO (SNP)” change to “the exogenous donors NO” (SNP)

*L25: “Pn and rETR”. Review the manuscript and define the abbreviations the first time they appear. Example

*The table feet of table 1, 2 and table 3 are the same, in 2, 3 simply indicating this fact is sufficient.

*L” Figure 3.” Figure 4 and throughout the paragraph.

*Throughout the manuscript there are different sizes of letters, please check

Round 2

Reviewer 2 Report

The authors have responded correctly to most of my suggestions and I accept the paper in its current version.